# Research on Routing Algorithm of Construction Robot Cluster Enhanced Ad Hoc Network

**DOI:** 10.3390/s23104754

**Published:** 2023-05-15

**Authors:** Ningke Wei, Liang Zhao, Dengfeng Chen

**Affiliations:** College of Information and Control Engineering, Xi’an University of Architecture and Technology, Xi’an 710055, China; zhaoliang@xauat.edu.cn (L.Z.); chdengf@xauat.edu.cn (D.C.)

**Keywords:** ad hoc network, connectivity, construction robot, link quality

## Abstract

An enhanced self-assembling network routing algorithm is proposed for the problem of weak connectivity of communication networks caused by factors such as movement or environmental interference in the construction and operation, and the maintenance of construction robot clusters. Firstly, the dynamic forwarding probability is calculated based on the contribution of nodes joining routing paths to network connectivity, and the robust connectivity of the network is achieved by introducing the connectivity feedback mechanism; secondly, the influence of link quality evaluation index Q balanced hop count, residual energy, and load on link stability is used to select appropriate neighbors for nodes as the subsequent hop nodes; finally, the dynamic characteristics of nodes are combined with the topology control technology to eliminate low-quality links and optimize the topology by link maintenance time prediction and to set robot node priority. The simulation results show that the proposed algorithm can guarantee a network connectivity rate above 97% under heavy load, reduce the end-to-end delay, and improve the network survival time, providing a theoretical basis for achieving stable and reliable interconnection between building robot nodes.

## 1. Introduction

Construction robots have thrived in the intelligent upgrading and transformation of the construction industry [1]; however, the complex construction site environment makes it difficult for traditional communication methods to guarantee the high-reliability interconnection communication needed for the coordinated control and scheduling of construction robots. There is an urgent need for a high-efficiency, high-quality, and low-cost communication network connection method. Wireless ad hoc networks are favored for their characteristics of no fixed infrastructure support and distributed communication [2].

Because of their relatively strong mobility and limited energy, the nodes of construction robots have high requirements for the effectiveness of routing protocols. At present, the design and optimization of routing protocols are mainly divided into four categories [3,4,5]: (1) Reducing node energy consumption; (2) Reducing node load; (3) Maintaining link stability; (4) Dynamic maintenance [6,7]. Building a high-reliability, large-bandwidth, and low-latency communication network required for coordinated control and scheduling for construction robot clusters [8,9] expounded the construction and spatial modeling of the ground robot cluster communication network and integrated positioning and communication into the robot network, but ignored the complex secondary, coupling superposition characteristics of the industrial site. In [10], potential fields and graph theory were introduced to optimize the link capacity of the self-assembled network of clustered robots. However, the network’s connectivity could not be guaranteed when there were more nodes and convergence was not fully achieved, leading to limitations in the scalability of the network.

### 1.1. Related Work

In terms of reducing node energy consumption, in [11], the energy efficiency of nodes is improved through clustering and efficient routing is established through the number of node hops, which has a good effect on the transmission rate of data packets. The research is based on static networks and considers the dynamic uncertainty of the scene less. In [12], the routing algorithm based on the Firefly algorithm to optimize FCM (Firefly algorithm to optimize fuzzy C-means, FFACM) establishes a cost function by calculating the link cost between nodes, which improves the energy consumption of nodes. In addition, the routing overhead can also be reduced by probabilistic broadcasting based on trust [13]. It can solve the broadcast storm phenomenon caused by many broadcast routing requests to a certain extent. It should be noted that in [13], the scene with uneven distribution of nodes is not considered.

In terms of reducing node load, in [14], the routing strategy of the ad hoc on-demand multipath distance vector (AODMV) protocol was improved, and the nodes with smaller MAC layer interface queue length and higher residual energy were selected to forward data packets. It was improved, but it is easy to initiate route discovery frequently. In [15], the hybrid routing algorithm integrating Ant Colony Optimization (ACO) and the minimum hop mechanism had good results in finding the optimal path, balancing network load, and maintaining network topology, but the algorithm added complexity. In [16], the channel busyness-based multipath load balancing routing protocol (CBMLB) distributed the load among available alternative paths to reduce a load of intermediate nodes with increased channel busyness. However, it relies too much on disjoint paths with lower congestion, which makes the network load unbalanced.

In terms of maintaining the stability of routing links, in [17], a more stable and reliable multi-path quality of service multicast routing protocol (SR-MQMR) considering stability and reliability was proposed for the bandwidth limitation of wireless nodes and the lack of sufficient multicast trees. According to the node signal strength one must select more stable routing nodes, but when the threshold for selecting stable nodes is equal to the average variance of neighbor node stability, this rule is unsuitable for selecting stable nodes. In [18], the link stability problem was described as the minimum cost capacity flow problem in routing selection, and the routing problem was solved in the minimum time, employing incremental subcontracting. However, there is no alternative route, which reduces the network’s reliability. In [19], a delay and link stability aware (DLSA) routing protocol was proposed to improve link stability by maximizing packet delivery rate and minimizing end-to-end delay but increasing the node load to some extent.

In terms of dynamic routing maintenance, in [20], the routing interaction process was maintained by managing the data packets and routing between the source and destination nodes. However, it still needs to be improved in terms of security. In [21], a routing selection method based on maintaining network connectivity to improve the overall life of the network was proposed. The network survival time, packet delivery rate, and network connectivity rate were improved through active and reactive routing maintenance strategies. However, the performance in the highly dynamic environment needs to be further verified. In order to avoid problems such as network connectivity failures caused by dynamic uncertainties, such as rapid network topology changes [22], based on node power control [23], the use of inter-node transmission power maximization will quickly generate dense network areas, making the network performance decrease [24,25], increasing the packet loss rate of network communication and reducing network throughput [26,27,28].

### 1.2. Motivations, Contributions, and Limitations

In the construction industry, the application of construction robots is gradually becoming popular [29]. Cluster construction robots will no longer be satisfied with independently completing a particular part of the work. Instead, it is necessary to jointly complete industrial tasks through coordinated control and achieve the goal of improving work efficiency [30]. With the continuous development of the intelligent process, people’s demand for mobile communication is not limited to daily life, and the demand for work is also becoming stronger and stronger. The characteristics of the distribution of construction robot clusters in industrial sites meet the requirements of distributed networking, so the research on construction robot cluster networking aligns with the current social science and technology development trend. However, it should be noted that the development and application of construction robots is increasing. There is an urgent need for coordinated communication among multiple robots. The complex environment of the construction site has seriously affected the connectivity of the construction robot ad hoc network [31]. The complex environment refers to the interference caused by metal building materials and walls on the construction industry site to the communication between construction robots. For the current situation of weak connectivity of coordinated communication between building robots, in this case, this paper proposes an enhanced ad hoc network algorithm (Ad hoc Cluster Enhanced Routing, ACER) for construction robot clusters. Firstly, by integrating the connectivity rate feedback mechanism and the routing path quality index to improve the routing path quality and network reliability, compared with other algorithms, this algorithm has a more significant effect in the construction environment, which significantly interferes with node communication. Regarding coverage area, the movement of construction robot nodes is adjusted based on the task priority, which reduces the impact of node movement on network coverage, further improves network connectivity, and improves the stability of the communication network of cluster construction robots. Finally, this was verified by simulation experiments. The validity and feasibility of the proposed algorithm are presented. The main contributions of this work are as follows:We designed a new route discovery method, which selects the relay nodes by calculating the node connectivity contribution to ensure the reliability of the communication network;We optimized routing path selection rules to quantify routing node performance;The topology control strategy based on link maintenance time and node priority was designed according to construction robots’ different mobility and construction sequence. The network’s connectivity in the construction environment is improved from the perspective of topology control, and the communication process between nodes is improved.

## 2. Problem Statement and System Model

### 2.1. Network Model

Represent the communication network by an undirected graph *G* = (*V*, *E*), where *V* = (*v*_1_,……,*v*_n_) denotes the set consisting of all nodes in the network and *E* denotes the set consisting of all link edges in the network, and *V*(*G*) and *E*(*G*) represent the set of vertices and the set of edges of *G*, respectively. Furthermore, make the following assumptions:*N* multi-class construction robot nodes equipped with airborne ad hoc network equipment are randomly located on each floor of a building, and each node has a unique ID;All construction robot nodes move randomly within the site according to their tasks and can only be active on the initial floor;Nodes contain the same amount of energy in the initial stage, can obtain the remaining energy at any time, and can interact with ID and location information between other nodes;The nodes can dynamically adjust the transmitting power according to the communication demand.

### 2.2. Node Movement Model

The wireless self-organizing network models were classified into individual and group mobility models according to the mobility mode [32]. Since the construction robots operate on different floor planes, the node individual movement model is designed to move only on the plane. Each node has an initial velocity and an initial direction, which are recalculated each time a new job is performed, and the node is updated with its velocity according to Equation (1):(1)$$v_{n}=a\times v_{0}+(1-a)\times v_{a}\times \sqrt{1-a^{2}}\times v$$
where *v*_0_ denotes the initial velocity, *a* is the parameter regulating the randomness, *v*_a_ denotes the average velocity, and *v* denotes the velocity parameter obeying a Gaussian distribution. The nodes are updated in the velocity direction according to Equation (2):(2)$$d_{n}=a\times d_{0}+(1-a)\times d_{a}+\sqrt{1-a^{2}}\times d$$
where is the *d*_0_ initial velocity direction, *d*_a_ is the average velocity direction, and *d* is the velocity direction parameter that obeys Gaussian distribution. The nodal coordinates of the node can be derived from the above Equation (3):(3)$$\left\{\begin{array}{c} X_{\mathrm{in}}=X_{i0}+v_{n}\times \cos(d_{n})\times t\\ Y_{\mathrm{in}}=Y_{i0}+v_{n}\times \sin(d_{n})\times t\\ Z_{\mathrm{in}}=Z_{\mathrm{in}} \end{array}\right.$$
where *X*_i0_, *Y*_i0_, and *Y*_i0_ denote the initial position coordinates of the node, and *X*_in_, *Y*_in_, and *Z*_in_ denote the position coordinates of the node after moving *t* moments.

### 2.3. Energy Consumption Model

In 3D wireless self-organizing networks, most of the energy consumed by nodes is used for communication, so the energy used by nodes for computation, storage, and sleep is negligible. Since many factors can interfere with node communication in practical application scenarios, a threshold *r*_*d*0_ was introduced in the energy consumption model to run different models, which is calculated as shown in Equation (4):(4)$$r_{d0}=\sqrt{\frac{\varepsilon_{fs}}{\varepsilon_{mp}}}$$
where *ε_fs_* denotes the free-space channel model signal amplifier power consumption and *ε_mp_* denotes the multi-path fading channel model signal amplifier power consumption, which are used in this paper to determine the selection thresholds for different channel models, taking the values of 10 × 10^−12^ J and 1.3 × 10^−15^ J, respectively. The energy consumed by a node in sending *k* bit data to a node *r_d_* apart is given in Equation (5):(5)$$E_{TX}(k,r_{d})=\left\{\begin{array}{l} E\times k+\varepsilon_{fs}\times k\times d^{2},r_{d}<r_{d0}\\ E\times k+\varepsilon_{mp}\times k\times d^{4},r_{d}>r_{d0} \end{array}\right.$$

When the distance *r_d_* between two nodes is less than the threshold *r*_*d*0_, the free-space propagation model is run, and vice versa, the multi-path fading channel model is run. The energy consumed by a node when it receives *k* bit data is shown in Equation (6):(6)$$E_{RX}(k)=E\times k$$
where *E* is the energy the node requires to send/receive a unit bit of data.

## 3. ACER Routing Algorithm

The ACER algorithm proposed in this paper aims to establish a connectivity-centered communication strategy to ensure the effectiveness of the communication process among construction robots through route selection and dynamic maintenance that incorporates routing path quality metrics. The algorithm is divided into four parts: route discovery, route selection, route maintenance, and topology control. The ACER routing algorithm flowchart is shown in Figure 1.

### 3.1. Route Discovery Based on the Connectivity Feedback Mechanism

The general case of route discovery is divided into four steps, which are: (1) the source node sends a route request information packet RREQ containing node identifier, node energy, and experience path to the neighbor nodes; (2) if the source node does not contain the target node in its neighbor node-set, the node in the neighbor node set continues to forward the node’s RREQ as a relay node; (3) the target node receives the RREQ from the node through one or more paths, establishes the set of reachable paths, and constructs route reply information packet RREP containing node identifier, node energy, experienced path, and link hops, and sends it to the source node along the original path; (4) after the source node receives the RREP from the target node by the specified time, it confirms the route reply information packet according to the agreed rules. It confirms the reachable path set, and then route discovery ends and enters route selection.

In the preceding step (2), the neighboring nodes of the source node tend to forward route requests as intermediate nodes to seek available routes. Broadcast routes and multicast routes tend to result in a large number of duplicate route requests being received by the forwarding node and the target node with high energy consumption. In contrast, single-route routing, such as distance vector routing algorithms, can forward data along a unique path to the destination node, but the complex field environment in dynamic networks makes a single routing path for which it is impossible to guarantee network connectivity. Probabilistic broadcasting, such as DV-cast (distributed vehicular broadcast-cast) protocol, has some shortcomings—the message cannot be propagated in two directions and the cached message is easily forwarded blindly. Therefore, the route discovery part of the routing algorithm in this paper introduces a connectivity feedback mechanism, i.e., the network connectivity contribution is calculated for the intermediate nodes that forward the route request data, and the probability of intermediate nodes forwarding the routing information is formed as shown in Equation (9) so that the forwarding nodes are selected. A comparison between the general route discovery process and the route discovery process in this paper is shown in Figure 2. In this paper, the route request RREQ message adds a connectivity state field in addition to the source node sequence number, destination node sequence number, node energy, and link quality, indicating the connectivity probability of the route path up to the current node.

The connectivity rate of the regional network centered on intermediate node *m* is calculated by constructing the adjacency matrix *A*. Suppose the adjacency matrix of intermediate node *m* is given in Equation (7):(7)$$A=\left[\begin{array}{cc} 1 & 1\\ 0 & 1 \end{array}\right]$$

The adjacency matrix characterizes the 1-hop pathway, and *A*^k^ can reflect the k-hop reachable situation. The matrix *P* was introduced as the connectivity judgment matrix, and *P_ij_* = 1 indicates that it is connected from *i* to *j*. Otherwise, it is not connected. The connectivity contribution is calculated as follows:Place the new matrix *P*: = *A*;*t* = 1, ……, *k*, *P* = *P*∗*A*, circular superposition to obtain the final connectivity matrix;Successive judgments are made on the *P* matrix *P_ij_*, and the number of elements with *P_ij_* > 0 is accumulated num.

The contribution of nodes as forwarding nodes to network connectivity *C* is obtained as shown in Equation (8):(8)$$C=\frac{num}{N(A)\times N(A)}$$
where *N*(*A*) denotes the dimension of the adjacency matrix *A*. The final probability that source node *i* selects intermediate node *m* as the forwarding node is obtained, as shown in Equation (9).
(9)$$P_{i,m}=\left\{\begin{array}{cc} C\times (1-e^{-\frac{\pi n}{NUM_{N(i)}}})\times \frac{d}{R} & ,m\in NUM_{N(i)}\\ 0 & otherwise \end{array}\right.$$
where *n* is the total number of nodes in the network, *NUM*_*N*(*i*)_ is the number of nodes in the node *i*’s neighbor node-set, *d* is the distance between source node *i* and intermediate node *m*, and *R* is the node communication radius. The improved route discovery algorithm is shown in Algorithm 1.
**Algorithm 1.** Routing discovery algorithm based on connectivity feedback mechanism.Input: source node *i*, Target Node *j*, adjacency matrix *A*, total number of nodes *n*, let *s* = 1;Output: forwarding nodes, forwarding probabilitybegininitializing the network; for *s* = 1: *n* update the set of neighbor nodes *N*(*i*) of source node *i*; end for if *m*
∈ *N*(*i*) calculate the node *m* connectivity contribution *C* using Equation (8); calculate the forwarding probability of node *m* using Equation (9); end ifoutput the node with the highest probability of forwarding and the probability value;End

### 3.2. Routing with Converged Routing Path Quality Metrics

When performing route discovery, there may be multiple paths. If the minimum hop count path is selected by simply comparing the path hop count hop, the average length of the routing path is often too long [33]. When neighboring nodes on the routing path are at the boundary of the maximum communication distance, subtle movement of nodes may lead to link breakage between nodes, so relying solely on hop count as routing path selection is likely to lead to the selection of paths with poor link quality, frequent initiation of route discovery, and degradation of network performance.

In evaluating the stability of the network routing path, when the nodes have sufficient energy, and the relative distance between nodes is less or much less than the communication distance, it indicates that the routing path has excellent stability, and the network topology is firmly connected. Therefore, a path quality evaluation metric *Q*_1_ between adjacent nodes and a multi-hop routing path quality evaluation metric *Q*_2_ was introduced to constrain the routing path selection mechanism and optimize the network connectivity. The path quality evaluation metric *Q*_1_ uses the relative distance between neighboring nodes to represent the stability of data transmission between neighboring nodes; the multi-hop routing path quality evaluation metric *Q*_2_ incorporates the number of routing path hops and node energy to characterize the link stability. The stability of a routing link largely determines the effectiveness of the communication network. When using the routing path quality evaluation metric Q to quantify the stability of a routing link, its value is between 0 and 1, and a larger value is more helpful for network connectivity. In this paper, the link that better ensures stable data transmission between nodes is selected according to its value.

The routing path quality evaluation metric *Q*_1_ determined by the relative distance is shown in Equation (10):(10)$$Q_{1}=\left\{\begin{array}{cc} 1 & ,0\leq d\leq \frac{R}{10}\\ \frac{1}{10}\times (ln{\left(\frac{R}{100\times d}\right)}^{2}+10) & ,\frac{R}{10}<d\leq R\\ \begin{array}{l} 0 \end{array} & \begin{array}{l} ,d>R \end{array} \end{array}\right.$$
where *R* denotes the node communication distance and d denotes the inter-node distance. The routing path quality evaluation metric *Q*_2_ determined by the number of path routing hops, node energy, and load is shown in Equation (11):(11)$$Q_{2}=\frac{1}{hop_{ij}}\times \frac{E_{i}-E_{u}}{E_{0}-E_{u}}\times \frac{C_{i}}{C_{0}}$$
where *hop_ij_* denotes the number of hops of the routing path from node *i* to node *j*, *E_i_* denotes the current remaining energy of node *i*, *E*_0_ denotes the maximum energy of node, and *E_u_* denotes the energy threshold of the node. When the remaining energy of the node is below this threshold, the node exits route discovery, i.e., the node is invalidated, where the energy threshold *E_u_* is defined as the energy required by the node to send one packet to a one-hop neighbor node, *C_i_* denotes the current amount of data cached by the node, and *C*_0_ denotes the cache capacity.

In Equation (10), the smaller the relative distance between two nodes, the larger the routing path quality evaluation index *Q*_1_, indicating that the routing path has better stability and more vital network connectivity, and when the relative distance *d* between nodes is greater than the node communication distance *R*, a direct link cannot be established between two nodes and the stability is expressed as 0. In Equation (11), a larger *Q*_2_ indicates that the source node has a better ability to complete data transmission, described explicitly as fewer hops between nodes, less energy consumption, and a more balanced residual energy and congestion level of nodes, indicating that the current link has better stability and reliability and better network connectivity.

### 3.3. Dynamic Routing Maintenance

It is discarded when a node repeatedly receives a routing message in the route establishment time. Otherwise, it is added, and the routing table is updated. The specific time measure is set to *T*, as shown in Equation (12):(12)$$T=\frac{6}{n}\sum_{0}^{n}t_{ij}$$
where *t_ij_* denotes the time required for any two nodes in the mobile self-assembly network to establish a route, *n* is the total number of nodes, and to ensure the accuracy of *T*, it is necessary to select n as many nodes as possible. Once the routing path is established, all the routing information received by the non-routing path is deleted; when the routing path is established, after *T* time, and the routing path still has not established communication, then the routing path is invalid, all the previously established routing paths are disconnected, and the paths are released. Two conditions trigger route maintenance: node movement and adding or removing nodes within the network.

The routing maintenance steps are as follows:Node *i* sends link confirmation information to neighboring nodes in the routing table and retains the node’s routing information if the return information is received within the cycle time. Otherwise, it is deleted, and the routing table is updated;Node *i* broadcasts a connection request packet with maximum power across the network. If it receives a reply packet from a node other than the node described in step 1, it updates the routing table by adding the routing information arriving at that node to the routing table.

If a node moves during the transmission of a packet from source node *i* to target node *j*, causing a topology change that results in a break in an otherwise valid link, node *m* at the path break returns a routing error message RERR containing information about the node at the link break to the previous hop node *l*. At this point, node *l* traverses the routing table to seek a new path to the target node *j*. If there is one, the original routing table contains nodes in the path from node *i* to node l that are updated in the routing table; if not, node *l* continues to forward RERR to the next hop node until it finds an available route. If RERR is forwarded to source node *i* and no available route is found, the route discovery is re-run.

### 3.4. Topology Control Based on Link Maintenance Time and Node Priority

Firstly, a topology prediction method based on the current position of nodes was introduced. For links between neighboring nodes, a link sustainability time was introduced to determine whether the link can remain connected in the next movement phase of the node [34]. As shown in Figure 3, the positions of node *d* and node *f* at moment *t* are (*x_d_*, *y_d_*, *z_d_*) and (*x_f_*, *y_f_*, *z_f_*), the velocities are *v_d_* and *v_f_*, and the velocity directions are *θ_d_* and *θ_f_*, respectively, and *R* is the communication radius of the nodes. Since the nodes still need to continue to move, the link maintenance time *T_L_* is defined, and *T_L_* can be deduced from the position relationship of the nodes as:(13)$$R^{2}={\left(a_{1}T_{L}{}_{f}^{d}+a_{2}\right)}^{2}+{\left(b_{1}T_{L}{}_{f}^{d}+b_{2}\right)}^{2}+c_{2}{}^{2}$$
(14)$${T_{L}}_{f}^{d}=\frac{\sqrt{(a_{1}{}^{2}+b_{1}{}^{2})(R^{2}-c_{2}{}^{2})-{\left(a_{1}b_{2}-a_{2}b_{1}\right)}^{2}}-(a_{1}a_{2}+b_{1}b_{2})}{a_{1}{}^{2}+b_{1}{}^{2}}$$
where *a*_1_ = *v_a_*cos*θ_d_* − *v_f_*cos*θ_f_*, *a*_2_ = *x_d_* − *x_f_*, *b*_1_ = *v_d_*sin*θ_d_* − *v_f_*sin*θ_f_*, *b*_2_ = *y_d_* − *y_f_*, *c*_2_ = *z_d_* − *z_f_*. In the previous section, it was shown that the robot nodes move only in the initial plane, so when considering the direction of node movement, only the direction of node movement in the plane is considered, i.e., the nodes are projected on the same plane for movement direction adjustment, but the distance is still in the spatial environment.

Secondly, for the actual situation of the construction robot application scenario, the robots of different task forms are prioritized by combining the actual construction sequence to maintain the connectivity of the communication network by using the lower priority robots as the preferred relay nodes during the construction robot movement [35]. Nevertheless, the low-priority node needs to suspend its task and change its previous direction of movement to one that avoids link breakage. As shown in Figure 4a, the topology diagram of robot nodes at one moment is distributed with robot nodes of different priorities. The arrows indicate the direction of node movement, and the topology diagram in the next cycle without optimizing its topology is shown in Figure 4b, where *a*, *b*, *d* and *e* denote different priority nodes.

Figure 4b shows that node *e* breaks the link with node *b* in the next cycle, affecting network connectivity. In this paper, after introducing the task priority of the construction robot, the stability of the network link is effectively ensured by moving the nodes with low priority in the direction of avoiding link breakage, ensuring better network connectivity.

The mobile direction of the low-priority node after determining the priority and the improved topology are shown in Figure 5, where *θ* denotes the node movement direction. It can be seen that the low-priority node *d* maintains a reliable communication link between node *b* and node e as a relay node after adjusting the mobile direction.

## 4. Simulation Experiments

To verify the performance of the proposed ACER routing algorithm, the field environment was simulated, and the algorithm was simulated by setting up an application scenario. Each iteration represents one random movement of nodes. The performance was analyzed regarding link quality, network connectivity, and the number of surviving nodes and was compared with the FFACM algorithm and the algorithm in the literature [36]. The simulation hardware equipment is as follows:CPU processor: i5-10400F @ 2.90 GHz;RAM memory: 8 GB;Hard disk: 1 TB;Software platform: NS 2.

To ensure the simulation results’ validity, each index’s results are the average of ten simulations. The main simulation parameters are shown in Table 1.

Initially, 100 network nodes were generated for simulation in the 3D scene, as shown in Figure 6. The four different colors and shapes of nodes represent nodes randomly distributed on four different horizontal floors. The *Z*-coordinates of the nodes are 75, 150, 225, and 300, respectively, to simulate the nodes at different floor heights. There are 25 priority 0, priority 1, priority 2, and priority 3 nodes, and the average number of neighboring nodes is about 8. Communication with nodes not in the neighboring node-set must be done through a multi-hop link. The test scenario does not have obstacles, and the regional distribution density of nodes on each floor is shown in Figure 7. In this thesis, the MAC and IP layers are fused to give each node a fixed and unique identity. The initial node sends connection requests to other nodes in the network with maximum power and establishes connections. The original topology diagram and the topology diagram, after running the topology optimization algorithm, are shown in Figure 8.

Figure 7 shows the distribution density of construction robot nodes on different floors in a 300 × 300 area. The nodes on each floor are not uniformly distributed but are more scattered in the application scenario based on their task requirements. From the comparison of the two figures in Figure 8a,b, we know that after running the topology optimization algorithm, the links with quality below the threshold are eliminated, the topology redundancy is reduced, and the network performance after topology sparse is subsequently investigated.

The connectivity rate is characterized by the ratio of the number of nodes in the maximum connected subgraph to the total number of nodes in the current network. As the number of nodes increases from 10 to 100, the connectivity rate curves of the ACER algorithm and the other two algorithms are derived for different numbers of nodes, as shown in Figure 9. The change curve of the connectivity rate can be described as follows, as the network scale continues to expand and the network connectivity rate continues to decrease. Finally, the connectivity rate of the FFACM algorithm and reference [36] algorithm were 93% and 92%, respectively. In comparison, the connectivity rate of the ACER algorithm reaches 97%. The algorithm connectivity rate is high and more stable, which is more suitable for the self-assembling network scenario of clustered construction robots.

The variation in the average link quality is shown in Figure 10. The ACER algorithm improves the topology for link maintenance time and node priority. The routing path quality index is integrated with the routing selection to select more stable and effective links. In the link quality comparison, the ACER algorithm is better than the FFACM and reference [36] algorithms. To verify the effectiveness of the ACER routing algorithm, the following analysis was performed in terms of the number of surviving nodes at different movement speeds of the construction robot nodes.

From Figure 11, it can be seen that the robot nodes survived more nodes per round at 0.5 m/s moving speed, and there were still 38 nodes surviving at 1000 rounds, two nodes surviving at 1000 rounds at 1.0 m/s moving speed, and all nodes died in the 631st and 873rd rounds at 1.5 m/s and 2.0 m/s moving speed, respectively.

A comparison experiment was conducted with the other two algorithms at a node movement speed of 0.5 m/s, as shown in Figure 12. The curve can be described as the gradual failure of nodes due to energy depletion as the number of iterations increases. The first node death rounds of this paper algorithm, FFACM algorithm, and reference [36] algorithm were 906, 305, and 458, respectively, while the last node death rounds were 1830, 1490, and 1620, respectively, and the ACER algorithm improves 22.8% and 13% in algorithm performance compared to the FFACM algorithm and literature [36] algorithm, respectively, effectively extending the network life cycle.

To verify the performance of the algorithm in this paper in a highly dynamic environment, the end-to-end delay at different packet rates was designed, as shown in Figure 13. End-to-end delay determines the adaptability of communication networks to topology changes. Although there is no clear end-to-end delay standard for self-assembled networks in industrial construction scenarios, the lower the latency, the more efficient and secure the communication network can be. The nodes move at the speed mentioned above of 0.5 m/s, and 4000 bit fixed-length packets were used for simulated communication, with each CBR (Constant Bit-Rate) source sending packets at rates ranging from 1 packet/s to 5 packet/s. As the CBR source sending packet rate increases, the end-to-end delay of the FFAMCM algorithm and reference [36] algorithm’s end-to-end delay in this paper also increases, which is due to the increase in packet rate that increases the network load. However, compared to the other two algorithms, this algorithm has a better performance. The algorithm in this paper has a stable performance because it combines the dynamic characteristics of nodes in the routing phase and considers the effects of load, residual energy, and hop count on routing, i.e., the subsequent hop nodes were selected with slower movement, lower congestion, higher energy, and fewer hops.

Based on the previous experiments, the end-to-end time delay comparison experiments were set up to verify the algorithm’s performance under different periods based on the characteristics of construction robots, which need to stay in different locations to complete related tasks such as construction, as shown in Figure 14. The nodes move at 0.5 m/s with 4000 bit fixed-length packets, the CBR source sends packets at 1 packet/s, and the simulation time is 400 s. As the pause time increases, the end-to-end delay of the algorithm in this paper, the FFACM algorithm, and the algorithm in reference [36] decrease continuously. However, the algorithm in this paper is significantly better than the other two algorithms, the increase in pause time causes node mobility to decrease, and the previous content has shown that the routing algorithm in this paper integrates several factors in routing node selection, so it is more adaptable to the dynamic changes of self-organized networks.

## 5. Conclusions

In order to improve the connectivity in the process of building self-organized communication networks among construction robots, an enhanced self-organized network routing algorithm for construction robot clusters was proposed. A topology optimization algorithm based on location prediction and robot node priority was proposed to address the problem of broken network routing paths due to complex environments and unstable links. The routing path establishment process for construction robot application environments was studied. The main results of the research work in this paper are as follows:A self-organizing network routing algorithm was designed with the connectivity rate as the core index, which facilitates the selection of optimal paths in the route selection process by adding a connectivity rate feedback mechanism to the route discovery process;Convergent routing path quality metrics quantitatively represent the link state of a routing path at a certain point in time and further optimize the path selection mechanism based on the node’s energy state, balancing relative distance, link hops, remaining energy, and load, so that the network eventually achieves a connectivity rate of 97% at 100 nodes;The topology prediction was also carried out based on the current position of the nodes and the direction of the moving speed, and the links with poor quality were eliminated. In contrast, the topology and communication strategy were further optimized by setting the priority of the construction robot nodes.

The simulation results also show that the algorithm can effectively improve the link quality and network connectivity and improve the self-adaptability and reliability of routing protocols, which performed better than the other two algorithms, and has specific significance for building self-organized networks in construction robotics application scenarios.

## Figures and Tables

**Figure 1 sensors-23-04754-f001:**
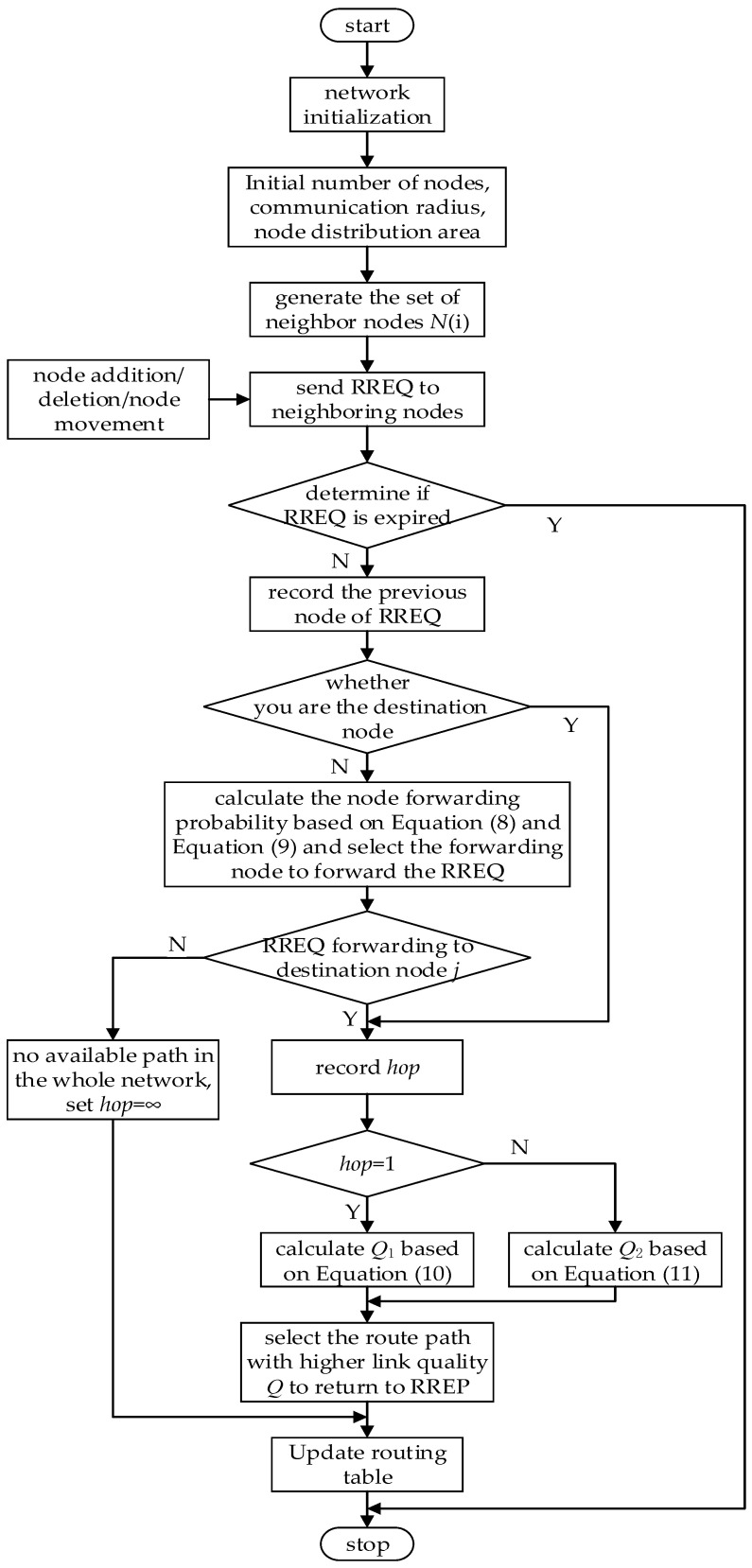
Flowchart of ACER routing algorithm.

**Figure 2 sensors-23-04754-f002:**
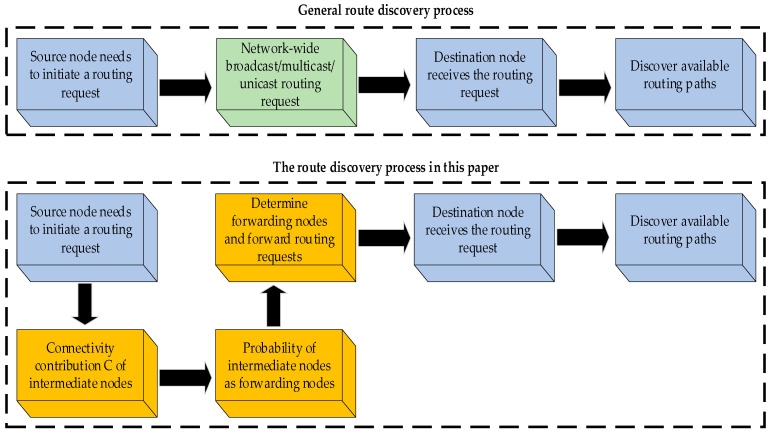
Comparison of route discovery processes.

**Figure 3 sensors-23-04754-f003:**
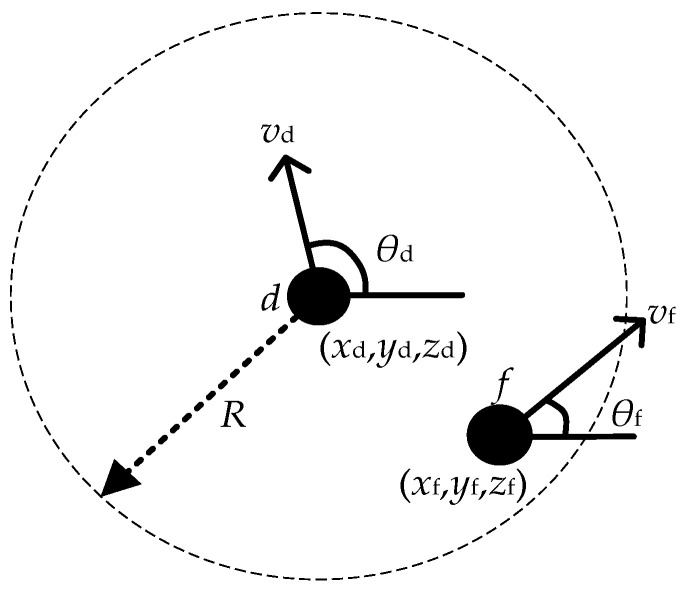
Two-dimensional projection of the relative positions of the nodes.

**Figure 4 sensors-23-04754-f004:**
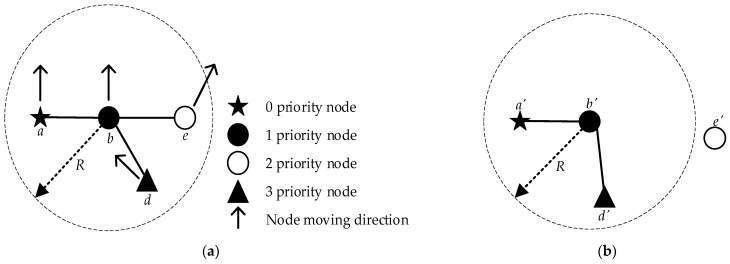
Network topology diagram. (**a**) Topology diagram at a point in the initial phase; (**b**) Topology diagram without topology control.

**Figure 5 sensors-23-04754-f005:**
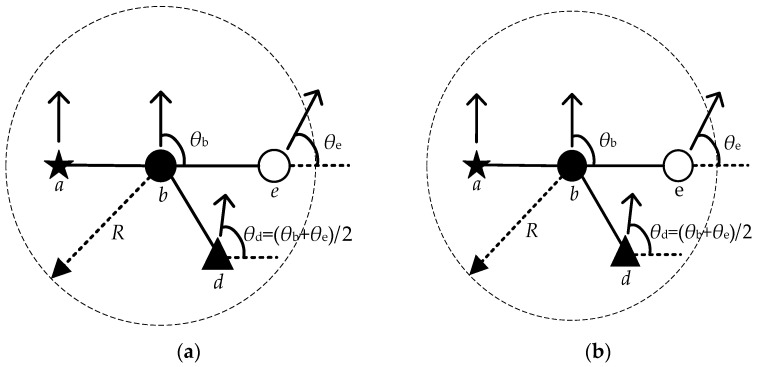
Network topology diagram. (**a**) Determining the direction of node movement after prioritization; (**b**) Final network topology.

**Figure 6 sensors-23-04754-f006:**
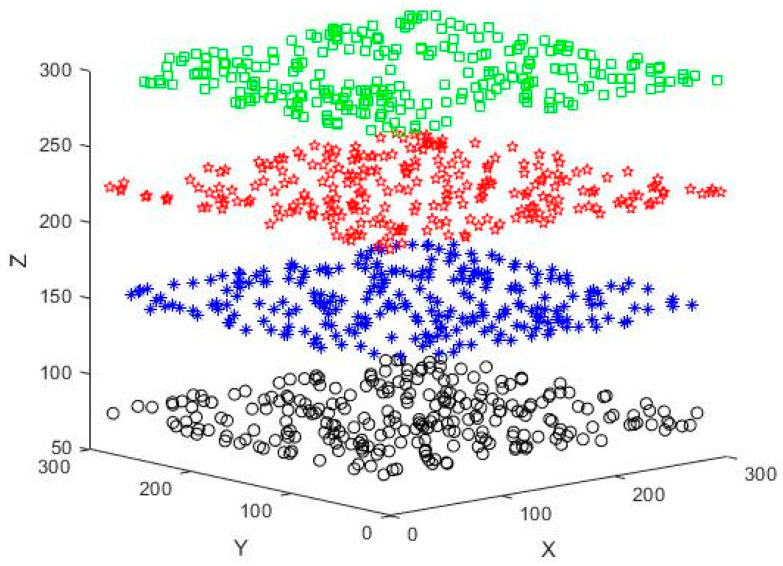
Node distribution diagram.

**Figure 7 sensors-23-04754-f007:**
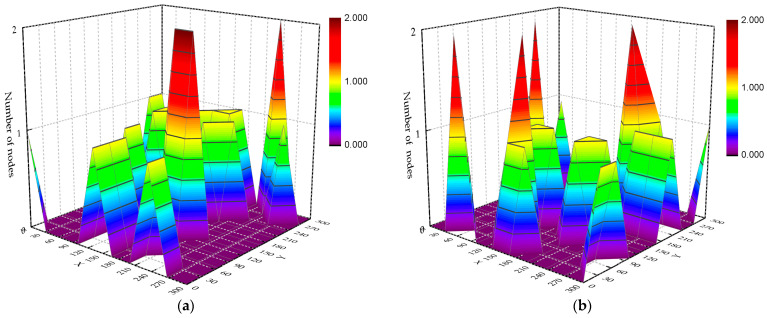
Distribution density of nodes on different floors: (**a**) Z = 75; (**b**) Z = 150; (**c**) Z = 225; (**d**) Z = 300.

**Figure 8 sensors-23-04754-f008:**
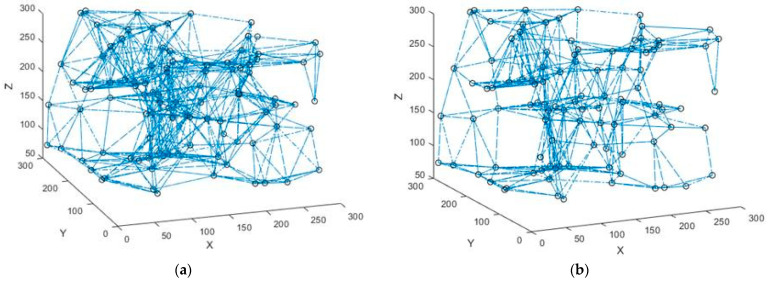
Network topology diagram. (**a**) Original topology of network nodes; (**b**) Optimized topology diagram.

**Figure 9 sensors-23-04754-f009:**
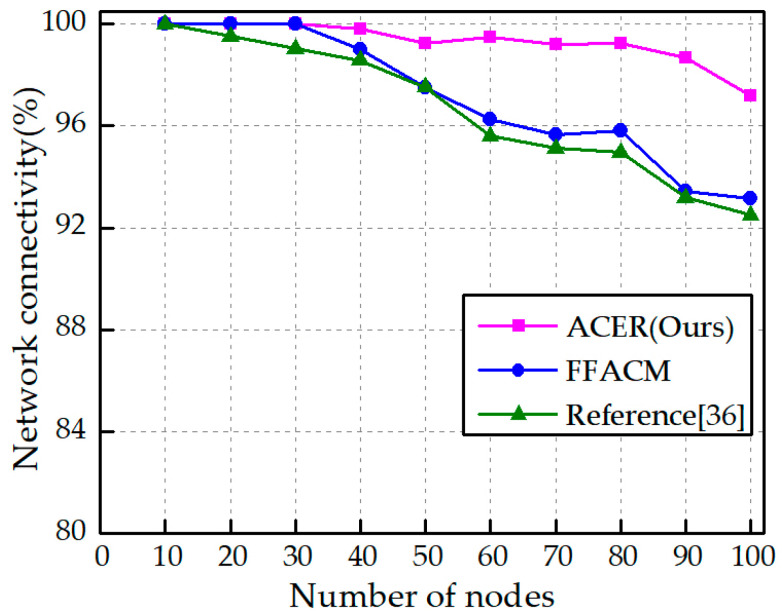
Relationship between the network connectivity and number of nodes [36].

**Figure 10 sensors-23-04754-f010:**
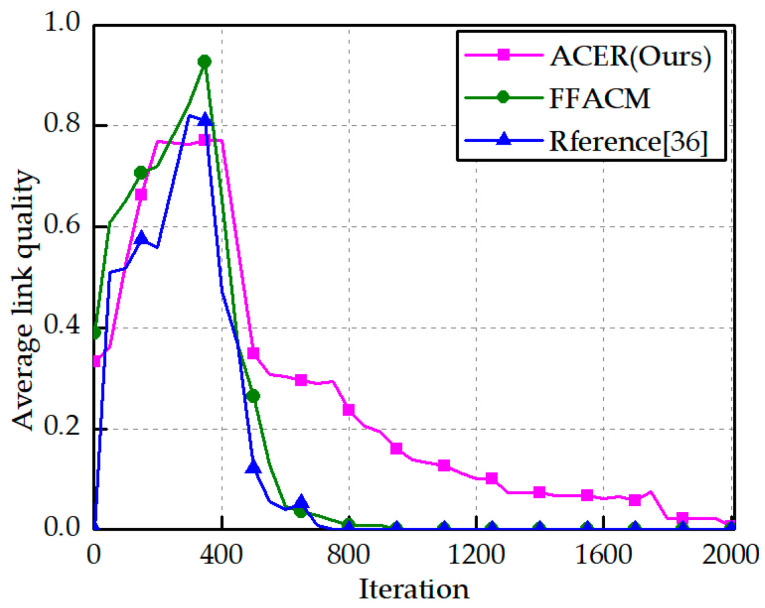
Average link quality [36].

**Figure 11 sensors-23-04754-f011:**
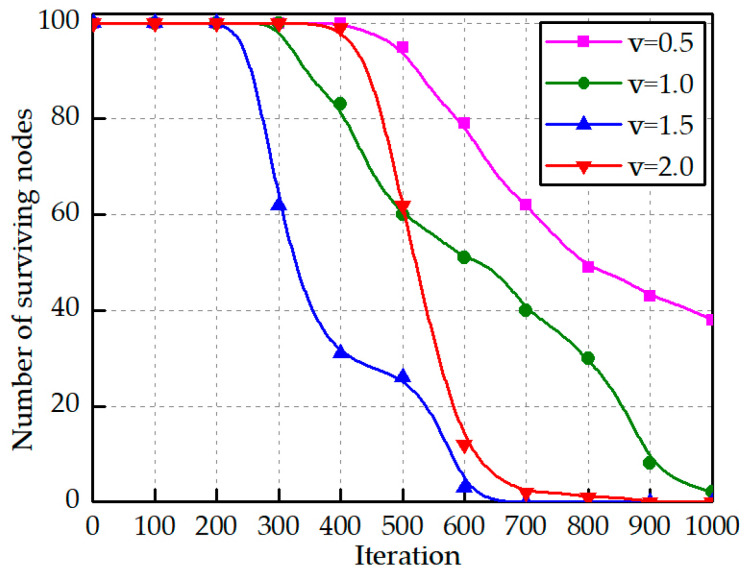
Number of surviving nodes per round at different moving speeds.

**Figure 12 sensors-23-04754-f012:**
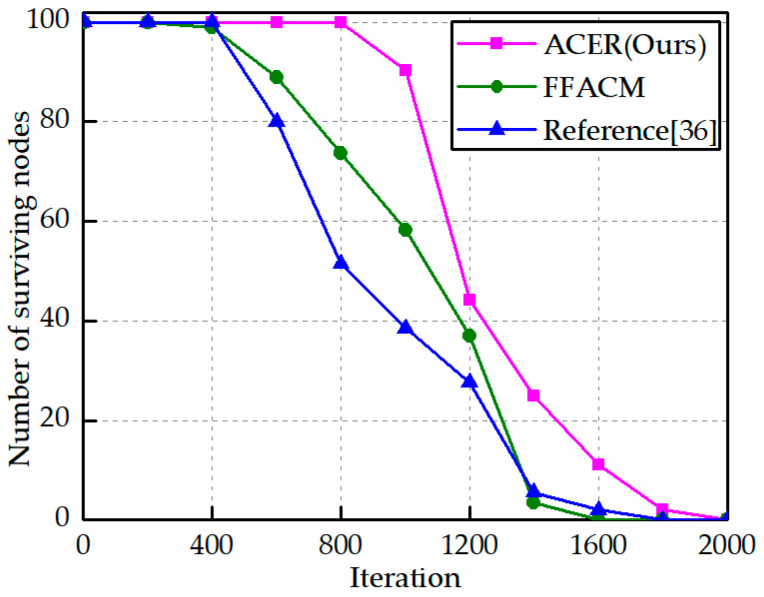
Network lifetime [36].

**Figure 13 sensors-23-04754-f013:**
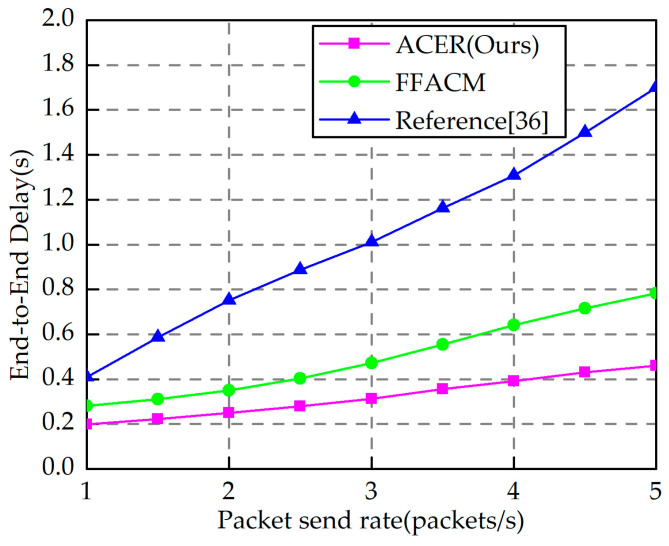
End-to-end delay at different packet send rates [36].

**Figure 14 sensors-23-04754-f014:**
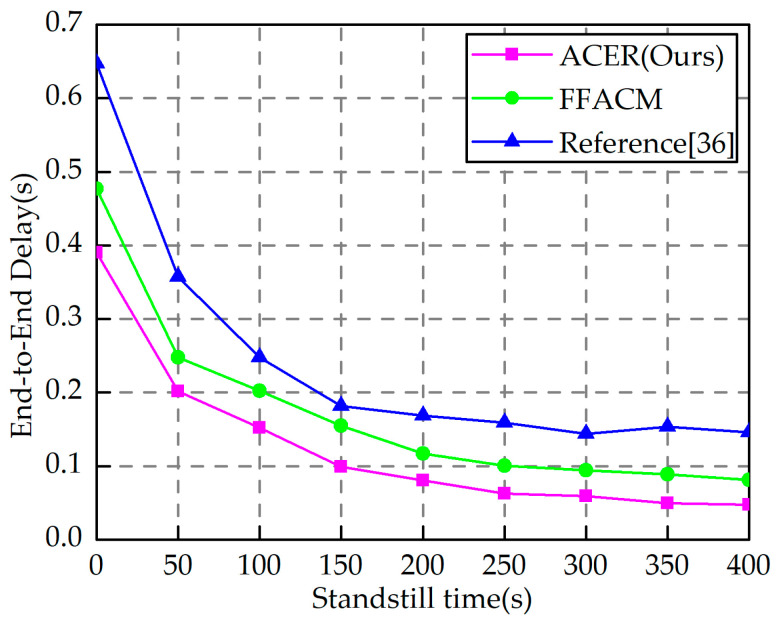
End-to-end delay at different standstill times [36].

**Table 1 sensors-23-04754-t001:** Simulation parameter setting.

Node distribution area	300 × 300 × 300 m
Number of mobile nodes	100
Initial energy of the node	1 J
Node communication radius	100 m
Maximum node movement speed	2 m/s
Maximum node dwell time	5 s
Unit packets	4000 bit
Network Protocols	IEEE 802.11

## Data Availability

Not applicable.

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
