# Peer review of "Research on Routing Algorithm of Construction Robot Cluster Enhanced Ad Hoc Network"

_sensors, 2023, doi:10.3390/s23104754_

Round 1

Reviewer 1 Report

In the peer-reviewed manuscript, the authors propose an improved self-organized network routing algorithm for clusters of construction robots. A topology optimization algorithm based on location prediction and robot node priority is proposed to address the problem of broken network routing paths due to complex environments and unstable links. The process of creating a routing path for application environments of construction robots is studied in depth.

The simulation results show that the proposed algorithm can guarantee the network connectivity rate above 97% under high load and improve the network survival time, which provides some theoretical basis to achieve stable and reliable connection between construction robot nodes.

The paper brings original novel information in the domain of the journal’s thematic focus. The research results are clearly distinguished from results adopted and used literary resources are mentioned properly. Credibility of published results is documented (experiments - simulations). Text readability and its linguistic correctness (even English texts, especially in the case of the technical terminology) is on the appropriate level. The graphic and content level of the individual chapters is appropriate and balanced.

I have several comments on the content of the article:

1. the abstract needs to be adjusted: clearly distinguish what is the goal and what is the benefit of the work !!!,

2. keywords need to be arranged alphabetically,

3. check the grammar, there are mistakes here, e.g. lines 150, 153, 156, 164, etc. not "Where" but "where"; or line 25 not "industry[1]." but " industry [1].", etc..

In general, after appropriate corrections and additions, I approve publication of this manuscript.

Check the grammar, there are mistakes here, e.g. lines 150, 153, 156, 164, etc. not "Where" but "where"; or line 25 not "industry[1]." but " industry [1].", etc..

Author Response

The authors would like to thank the reviewers for their comments. The paper was revised based on the reviewers’ comments and all mentioned changes are highlighted in the revised version of the paper.

Comment 1: The abstract needs to be adjusted: clearly distinguish what is the goal and what is the benefit of the work.

Response 1: Thank you for your suggestion. We have reorganized the abstract to distinguish between the study objectives and the related research done to achieve them. We have revised it in the manuscript (abstract) and described it in red as follows:

Firstly, the dynamic forwarding probability is calculated based on the contribution of nodes joining routing paths to network connectivity, and the robust connectivity of the network is achieved by introducing the connectivity feedback mechanism; secondly, the influence of link quality evaluation index Q balanced hop count, residual energy, and load on link stability is used to select appropriate neighbors for nodes as the subsequent hop nodes; finally, the dynamic characteristics of nodes are combined with the topology control technology to eliminate low-quality links and optimize the topology by link maintenance time prediction and to set robot node priority.

Comment 2: Keywords need to be arranged alphabetically.

Response 2: Thank you for your suggestion. We have reordered the keywords and arranged them in alphabetical order as follows:

Ad hoc network; Connectivity; Construction robot; Link quality

Comment 3: Check the grammar, there are mistakes here, e.g. lines 150, 153, 156, 164, etc. not "Where" but "where"; or line 25 not "industry[1]." but " industry [1].", etc..

Response 3: Thank you for your suggestion. We found some grammatical errors during our review of the written English, revised parts of the paper to lowercase the initials of the explanatory formulae, and corrected other grammatical errors, which are highlighted in the revised version of the paper.

Reviewer 2 Report

The authors present an interesting paper about a novel routing algorithm of construction robot cluster enhanced ad hoc network. However they should improve some issues, namely:

1.- it is advisable to carry out a review of the written english, as the document has some typos, and rewrite terms like "muscular mobility".

2.- During the presentation of the state of art, the authors refer to many acronyms, which deserved to be accompanied by the designation in full, to identify the different methods they use as a comparison. These are also, sometimes, characterized in a very redutive way in terms of advantages and disadvantages.

3.- The suthors' theory is essentially around the concept of the "connectivity", without however being very clear as to the parameters (energy, path link, latency) and their thrshoulds.

4. - About equation (4) it states that "efs and emp are communication energy parameters", is too short. What do these variables actually represent and how are they determined?

5.- Pg 5:Ln 185, replace "(2) after" by "(4) after"

6.- Also the test scenario should be better characterized. How many nodes are in/out of the communication radius? How many nodes have "low priority tasks"? Are there obstacles or not? what is the distribution density of the nodes?

7.- Although the conclusions claim that the ACER method performsbetter than the methods used, FFACM and Reference[36], nothing is referred to the temporal question, how does the proposed method behave in highly dynamic environments? Does the temporal convergence of the proposed method (for ex. connectivity determination) meet the requirements of the time constant of the process in which it will be applied?

Finally, congratulation on your work

Author Response

The authors would like to thank the reviewers for their comments. The paper was revised based on the reviewers’ comments and all mentioned changes are highlighted in the revised version of the paper.

Comment 1: It is advisable to carry out a review of the written english, as the document has some typos, and rewrite terms like "muscular mobility".

Response 1: Thank you for your suggestion. As you mentioned, a number of errors were identified during our thorough review of the written English and are highlighted in the revised version of the paper.

Comment 2: During the presentation of the state of art, the authors refer to many acronyms, which deserved to be accompanied by the designation in full, to identify the different methods they use as a comparison. These are also, sometimes, characterized in a very redutive way in terms of advantages and disadvantages.

Response 2: Thank you for your suggestions. We have added the full names of the acronyms involved in the content of the manuscript, specifically AODMV, CBMLB, SR-MQMR, and DLSA in section 1.1, and also refined the strengths and weaknesses of the methods in reference [19][21].

Comment 3: The suthors' theory is essentially around the concept of the "connectivity", without however being very clear as to the parameters (energy, path link, latency) and their thrshoulds.

Response 3: Thank you for your suggestions. In order to better clarify the relationship between parameter path links, energy, delay and connectivity, and the use of their thresholds in this paper, firstly, the key role of the routing path quality evaluation metric Q for ensuring network connectivity is added in Section 3.2, and the range of its values and its use in this paper is explained; secondly, the explanation of Equation (11) illustrates that the node residual energy Ei below the energy threshold E0 will make the and the energy required to send/receive a unit packet under different channel models is illustrated in Section 2.3; finally, its significance and the impact of its different values on the communication network are illustrated in the new comparative experiment on end-to-end delay in Section 4.

Comment 4: About equation (4) it states that "efs and emp are communication energy parameters", is too short. What do these variables actually represent and how are they determined?

Response 4: Thank you for your suggestion. To better illustrate what the parameters εfs and εmp in Eq. (4) actually represent and the determination of their values, we have added the role of the parameters in this manuscript to the content of their explanation after Eq. (4), stating that efs and emp are the amplifier power consumption of the free-space model and the multipath fading channel model, respectively, and are used in the study to determine the threshold for selecting the channel model, and giving the threshold values in the study the actual values, which are 10 × 10-12 J and 1.3 × 10-15 J, respectively.

Comment 5: Pg 5:Ln 185, replace "(2) after" by "(4) after".

Response 5: Thank you for your suggestion. There was a writing error in the original line 185 and we have changed (2) after to (4) after.

Comment 6: Also the test scenario should be better characterized. How many nodes are in/out of the communication radius? How many nodes have "low priority tasks"? Are there obstacles or not? what is the distribution density of the nodes?

Response 6: Thank you for your suggestions. In order to better characterize the test scenario, we added descriptions about the number of nodes of different priority levels, obstacles, and a legend (Figure 7) to more visually illustrate the distribution density of nodes in different floor areas in Section 4. In addition, the number of nodes located inside/outside their communication radius varies for different nodes, so the average number of nodes in the set of network nodes its neighbors is calculated and used to characterize the number of nodes inside/outside the communication radius.

We have added to this in the manuscript (abstract) and depicted it in red as follows:

There are 25 priority 0, priority 1, priority 2, and priority 3 nodes, and the average number of neighboring nodes is about 8. Communication with nodes not in the neighboring node-set must be done through a multi-hop link. The test scenario does not have obstacles, and the regional distribution density of nodes on each floor is shown in Figure 7.

Figure 7 shows the distribution density of construction robot nodes on different floors in a 300×300 area. The nodes on each floor are not uniformly distributed but are more scattered in the application scenario based on their task requirements.

Comment 7: Although the conclusions claim that the ACER method performsbetter than the methods used, FFACM and Reference[36], nothing is referred to the temporal question, how does the proposed method behave in highly dynamic environments? Does the temporal convergence of the proposed method (for ex. connectivity determination) meet the requirements of the time constant of the process in which it will be applied?

Response 7: Thank you for your suggestions. We have added end-to-end delay comparisons with different packet rates and different pause times in the comparison experiments, as shown in Figure 13 and Figure 14, to verify the performance of the algorithm in this paper when the self-organized network environment changes dynamically. The simulation results show that the algorithm in this paper can better adapt to the heavy load and dynamic environment, and after the routing information changes due to the topology change, the algorithm in this paper can respond more quickly and establish a valid route to transmit the routing information, and the end-to-end delay is less than 0.4s, which can ensure the effectiveness of the communication process in the practical application process.

Round 2

Reviewer 2 Report

The authors generally implemented the recommendations and observations indicated, so I am of the opinion that the paper meets the conditions to be published.

Congratulations on your work.